# A Unified Conditional Framework for Diffusion-based Image Restoration

**Yi Zhang** [1]   **Xiaoyu Shi** [1]   **Dasong Li** [1]   **Xiaogang Wang** [1]   **Jian Wang** [2] *   **Hongsheng Li** [134] *

[1] CUHK MMLab    [2] Snap Research    [3] Centre for Perceptual and Interactive Intelligence
[4] Shanghai AI Laboratory

zhangyi@link.cuhk.edu.hk,   jwang4@snapchat.com,   hsli@ee.cuhk.edu.hk
https://zhangyi-3.github.io/project/UCDIR

## Abstract

Diffusion Probabilistic Models (DPMs) have recently shown remarkable performance in image generation tasks, which are capable of generating highly realistic images. When adopting DPMs for image restoration tasks, the crucial aspect lies in how to integrate the conditional information to guide the DPMs to generate accurate and natural output, which has been largely overlooked in existing works. In this paper, we present a unified conditional framework based on diffusion models for image restoration. We leverage a lightweight UNet to predict initial guidance and the diffusion model to learn the residual of the guidance. By carefully designing the basic module and integration module for the diffusion model block , we integrate the guidance and other auxiliary conditional information into every block of the diffusion model to achieve spatially-adaptive generation conditioning. To handle high-resolution images, we propose a simple yet effective inter-step patch-splitting strategy to produce arbitrary-resolution images without grid artifacts. We evaluate our conditional framework on three challenging tasks: extreme low-light denoising, deblurring, and JPEG restoration, demonstrating its significant improvements in perceptual quality and the generalization to restoration tasks.

## 1   Introduction

Image restoration tasks are ill-posed problems whose solutions are not unique. But, most existing methods for image restoration rely on learning regression models that produce deterministic results. Unfortunately, such approaches often learn the "averaged" results of potential ground truth distribution, leading to blurry outputs that are not consistent with human perception. With the development of Diffusion Probabilistic Models (DPMs), there exists remarkable progress in image generation, which is able to generate realistic images with fine details. This progress has shed light on diffusion-based image restoration tasks, which utilize conditional DPMs to generate images that align well with the corresponding ground truth. With the input degraded input, conditional DPMs can produce realistic images with significant improvements in perceptual quality.

When applying DPMs to image restoration, the crucial challenge lies in how to effectively integrate degraded images and other conditional information (*e.g.*, diffusion time step, degradation type, and strength) into the DPMs. A better conditional framework can greatly enhance the generative capacity of DPMs and guide them to generate realistic images that faithfully align with the ground truths. However, it has been mostly overlooked in existing diffusion-based methods [50, 42, 41, 12]. Typically, these methods simply concatenate the degraded image as the input to the diffusion model, which fails to fully exploit the potential of the diffusion model.

In this paper, we present a unified conditional framework for diffusion-based image restoration. It effectively integrates multiple sources of conditional information into blocks of DPMs. This

---

*Corresponding Author

37th Conference on Neural Information Processing Systems (NeurIPS 2023).

integration enables spatially adaptive conditioning over blocks of the diffusion models, resulting in perceptual quality improvements of restored images. As shown in Fig. 1 (left), the framework consists of two models: an initial predictor, implemented as a lightweight UNet, and a diffusion model. The initial predictor generates the initial restoration output, capturing the coarse structure and deterministic components of the final output. The diffusion model is then employed to be guided by the initial result to predict the residual of the initial guidance. Since the initial predictor restores the initial results of the final output, we leverage it as the spatial guidance to the block of DPMs. This spatial guidance, along with other scalar conditional information, is integrated into the diffusion model blocks, enabling precise and adaptive conditioning during the restoration process.

To enhance the handling of conditional information within each block, we propose an Adaptive Kernel Guidance Block (AKGB) that effectively fuses the conditional information into DPMs. The key idea is to generate adaptive kernels for each spatial location by fusing a set of learnable kernels based on conditional information. Specifically, we encode both spatial information and auxiliary scalar information into the feature maps. These two types of information are then fused as multi-source fusion weights. The fusion weights are used to linearly combine a set of learnable kernels, resulting in fused kernel weights for each position. This fusion process takes into account both the spatial and scalar conditional information, enabling each position within the diffusion model to have its own customized kernel weights. It efficiently injects the condition information into the diffusion model and enhances its spatial adaptability and flexibility. This approach allows the models to effectively utilize the conditional information, leading to improved performance in handling complex image restoration tasks.

In practical applications, another important problem is to adopt diffusion models to generate high-resolution images [41]. However, direct applying the patch-based method often introduces noticeable artifacts due to inconsistent results along the patch boundaries. Inspired by the properties of image denoising models, we propose an inter-step patch-splitting strategy to perform patch-splitting at each diffusion time step. This strategy achieves consistent and artifact-free results in the restoration process for arbitrary-resolution images.

We evaluate this framework on three challenging image restoration tasks: extreme low-light denoising, image deblurring, and JPEG restoration. The experiments demonstrate that our method generalizes well to all three tasks and, achieves significantly higher perceptual quality than strong regression baselines and recent diffusion-based models.

Our contributions can be summarized as follows:

- We propose a unified conditional framework for diffusion-based image restoration tasks. It leverages a UNet to predict the initial guidance and enable integrating the multi-sources conditional information to every block to better guide the generative model.

- To effectively incorporate conditional information into diffusion models, we design a basic module and an Adaptive Kernel Guidance Block (AKGB). It combines the spatial guidance and auxiliary scalar information to adaptively fuse the dynamic kernels in each diffusion model block.

- A simple yet effective inter-step patch-splitting strategy is proposed for handling high-resolution images in low-level vision tasks. This practical strategy enables diffusion models to generate consistent high-resolution images without grid artifacts.

- Through extensive experiments on extreme low-light denoising, image deblurring, and JPEG restoration tasks, we demonstrate that our method not only achieves a significantly higher perceptual quality than strong regression baselines and recent diffusion-based models but also show good generalization to various restoration tasks.

## 2 Related Work

Our method draws inspiration from both regression-based and generative methods in image restoration, as well as dynamic networks.

### 2.1 Generative Image Restoration

**Image restoration.** Image restoration has been a highly active research topic in computer vision for several years. Due to the ill-posed nature of the problem, each degraded image has multiple potential solutions. Traditional methods have employed various priors and degradation models [18, 56]

to mitigate the challenges in image restoration. Traditional methods often rely on priors such as sparsity [17, 32], non-local self-similarity [13, 5], and total variation [40] to regularize the restoration process. While traditional methods have made significant progress, they struggle with complex or challenging cases. With the rise of deep learning, learning-based methods have become the mainstream approach in image restoration [53, 8, 11, 29, 28, 1, 6, 36, 55, 30]. These methods leverage large-scale training datasets and deep neural networks to learn the mapping between degraded inputs and clean outputs. Learning-based methods have shown remarkable performance improvements by implicitly capturing complex image patterns and structures. While learning image restoration from a large dataset shows promising performance improvements, they can only produce deterministic results for this ill-posed problem.

**Generation-based methods.** To mitigate the limitations of predicting deterministic solutions, a promising avenue is the adoption of generation-based methods. Unlike traditional approaches, these methods generate a distribution for a given degraded input, offering a more flexible and probabilistic solution. One popular approach in generation-based image restoration is the use of generative adversarial networks (GANs). Although several GAN-based methods have been successfully applied to specific domains such as face restoration [44] or super-resolution [45], generating artifacts-free and realistic outputs for general image restoration remains a considerable challenge for GAN-based methods.

An emergent and promising direction for generation-based methods lies in the utilization of diffusion models. These models have showcased impressive performance in diverse domains such as image generation [39, 14, 20], image editing [19, 4], and super-resolution [42]. In image restoration, there are several methods have been proposed. S3 [42] proposes conditional Diffusion Probabilistic Models (DPMs) to handle image super-resolution. Then, it has been extended to other image restoration tasks [41]. For image deblurring, DvSR [50] proposes a residual learning framework and icDPM [38] improves the out-of-distribution performance through a multi-scale representation. Most existing works simply concatenate the conditional information into the diffusion models, which largely limits the generation ability of diffusion models.

In addition to supervised methods, there exist unsupervised approaches [26, 25, 48] that decompose the image representation from different perspectives and directly integrate the degraded image into a pretrained diffusion model. Although these methods eliminate the need for retraining, they often suffer from poorer performance compared to the task-specific training method.

## 2.2 Dynamic Networks in Image Restoration

The significance of adaptive processing in image restoration tasks, taking into account factors such as image texture, degradation types, and degradation strength, is widely acknowledged in the field. In line with this concept, several learning-based methods have been proposed to generate dynamic kernels [22] for image restoration.

One track of the work is to employ a network to predict the spatially invariant convolution kernels [2]. This approach aims to capture the varying characteristics of different image regions and adapt the convolution operations accordingly. Building upon this foundation, KPN [34] extends the kernel prediction network into the burst image denoising task. The predicted kernels are utilized to aggregate the input burst images directly, enabling improved denoising performance and better image texture preservation. Instead of predicting the dynamic kernel directly, BPN [51] reduces the complexity of the kernel prediction idea by decomposing the predicted kernel matrix. Malleable convolution [24] improves the efficiency of kernel prediction by producing low-resolution spatial kernels and interpolating them for each position. By incorporating adaptive processing techniques, these methods demonstrate the ability to effectively adapt to different image restoration scenarios, leading to enhanced restoration quality and performance.

Our proposed method shares a similar underlying principle with dynamic kernels, as it leverages the integration of multi-source conditional information. These include the degraded input, diffusion time step, and degradation information. By incorporating these factors, our method dynamically generates adaptive kernels that effectively handle feature maps within each block.

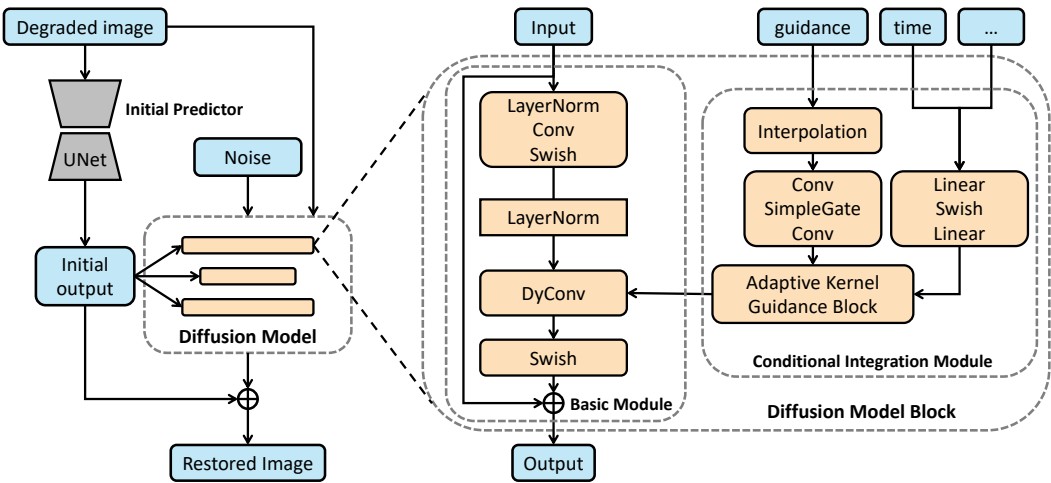

Figure 1: **Left**: An overview of the conditional framework. **Right**: The diffusion model block. "..." on the top-right represents the auxiliary scalar information for different tasks (*e.g.*, noise level, blur type.).

## 3 Method

### 3.1 Overview of Conditional Framework

Our objective is to design a unified conditional framework for image restoration tasks. The input conditional information for this framework consists of two components: the degraded image and auxiliary scalar information. The degraded image represents the image to be restored, while the auxiliary scalar information can include degradation types, strength, or other relevant details specific to each restoration task.

To enhance the integration of conditional information, we first employ a lightweight U-Net to predict an initial output as shown in Fig. 1 (left). This initial output captures the low-frequency and deterministic aspects of the final restored image, which are easier to restore and contain essential structural information. We utilize this initial output as the spatial guidance for the diffusion model. Together with the auxiliary scalar information (*e.g.*, degradation type, diffusion time step), we inject them into every block of the diffusion model, enabling better control and guidance for the diffusion models. This injection not only provides a comprehensive context but also enhances the flexibility of our framework. Following DvSR [50], we employ the diffusion model to capture the residual distribution of the initial output. Consequently, the training loss is modified as

$$\mathbb{E}_{(\boldsymbol{x},\boldsymbol{y})}\mathbb{E}_{\boldsymbol{\epsilon},\gamma}\left\|f_\theta(\sqrt{\gamma}(\boldsymbol{y}-u_\phi(x))+\sqrt{1-\gamma}\boldsymbol{\epsilon},\boldsymbol{x},\gamma)-\boldsymbol{\epsilon}\right\|, \tag{1}$$

where the noise vector $\epsilon \sim \mathcal{N}(0, I)$, $(\boldsymbol{x}, \boldsymbol{y})$ is the sampled degraded-clean image, $f_\theta$ denotes the diffusion model, $u_\phi$ represents the initial predictor, and $\gamma \sim p(\gamma)$ is a distribution related to the noise schedule. The highlight blue part in Eq. (1) shows the key modification of the "residual modeling".

### 3.2 Diffusion Model Block

**Basic Module**    In our approach, we have designed a basic module for the diffusion model used in image restoration tasks. We aim to keep the module as simple as possible by leveraging existing image restoration backbones. Instead of utilizing complicated operators, we try to make it as simple as possible by adopting existing image restoration backbones. For each block, we leverage two convolution layers. Before each of them, we introduce a LayerNorm to stabilize the training process [15]. Following the previous generative works [3, 42, 50], we employ the Swish as the activation function. A shortcut is applied to perform the residual learning. To enable the injection of the conditional information, the second convolutional kernels are designed to be dynamic based on the conditions.

**Conditional Injection Module**    To better integrate the conditional information into the block, we propose a Conditional Integration Module (CIM). Within CIM, the guidance is first scaled to match the resolution of the feature map within the block. This scaled guidance then passes through two

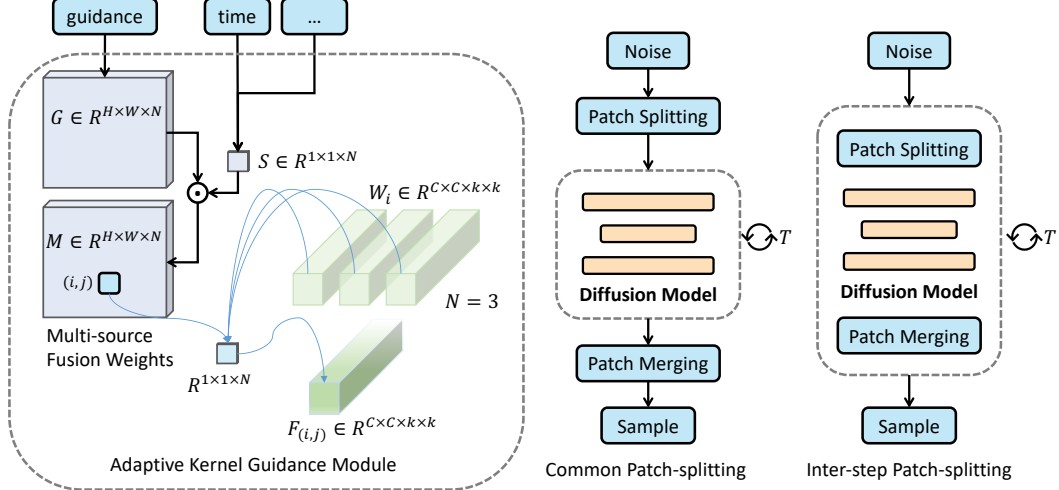

Figure 2: **Left**: Adaptive Kernel Guidance Module (AKGM). Here, $N$ is the number of kernel bases. We set $N = 3$ for visualization. **Right**: Inter-step patch-splitting.

convolution layers with a SimpleGate activation [8], effectively adjusting the channel number and producing the feature map $G$. Simultaneously, the auxiliary scalar information is processed through a two-linear-layer branch with Swish activation in between, resulting in the feature map $S$.

The feature maps $G$ and $S$ are subsequently passed to the Adaptive Kernel Guidance Module (AKGM) to generate dynamic kernels for the second convolution layer in the basic module, as depicted in Fig. 1. The key idea behind the AKGM is to adaptively fuse the kernel bases, allowing each spatial location to handle the feature map based on the fused multiple sources of conditional information. As shown in Fig. 2 (left), in each AKGM, we have $N$ learnable kernel bases denoted as $W_b \in \mathbb{R}^{C \times C \times k \times k}$, where $C$ represents the number of channels, and $k$ denotes the kernel size. These kernel bases are trained to handle different cases and scenarios. The scalar information (*e.g.*, time) is encoded by a two-linear-layer branch with Swish activation in between. The feature maps $G \in \mathbb{R}^{H \times W \times N}$ and $S \in \mathbb{R}^{1 \times 1 \times N}$ are fused using pointwise production to generate the multi-source fusion weights $M \in \mathbb{R}^{H \times W \times N}$. Here, $H$ and $W$ denote the height and width of the feature map, respectively, and $N$ represents the number of kernel bases. To obtain the fused kernel for a specific position $(i, j)$, denoted as $F_{(i,j)}$, it is obtained by linearly fusing the kernel bases according to the multi-source fusion weights at the same position. Formally, it can be expressed as

$$F_{i,j} = \Sigma_{b=0}^{N-1} M_{i,j}[b] W_b. \tag{2}$$

Fig. 2 illustrates the dynamic kernel generation process for a specific position $(i, j)$ with the number of kernel bases set to $N = 3$.

**Computational analysis.** In the standard convolution operation, given an input size of $H \times W \times C$, the computational cost is $HW(k^2 C^2)$ MACs. However, for the Adaptive Kernel Guidance Module (AKGM), calculating the dynamic kernel for each position introduces additional computational complexity. If we use the convolution kernels as the kernel bases, the computational cost is $HW(NC^2 k^2) + NC^2 k^2$, which increases linearly with the number of kernel bases. To reduce the computational burden, we adopt grouped convolution kernels with a group number of $N$ as the kernel bases. As a result, the computational cost is reduced to $HW(C^2 k^2) + C^2 k^2$. The core operation in AKGM has a comparable computational cost to a convolution layer. Besides, the additional workload introduced by the module is minimal, as the number of channels is set to a small number.

### 3.3 High-resolution Image Inference

In theory, constructing the diffusion model using pure convolution layers allows for arbitrary-size image inference. However, in practice, it is impossible to infer a very high-resolution image directly due to the limited GPU memory [41]. As shown in Fig. 2 (right), a common approach in image restoration is to split the input image into overlapping patches and independently perform inference on each patch. The output patches are then merged to reconstruct the entire image. However, when

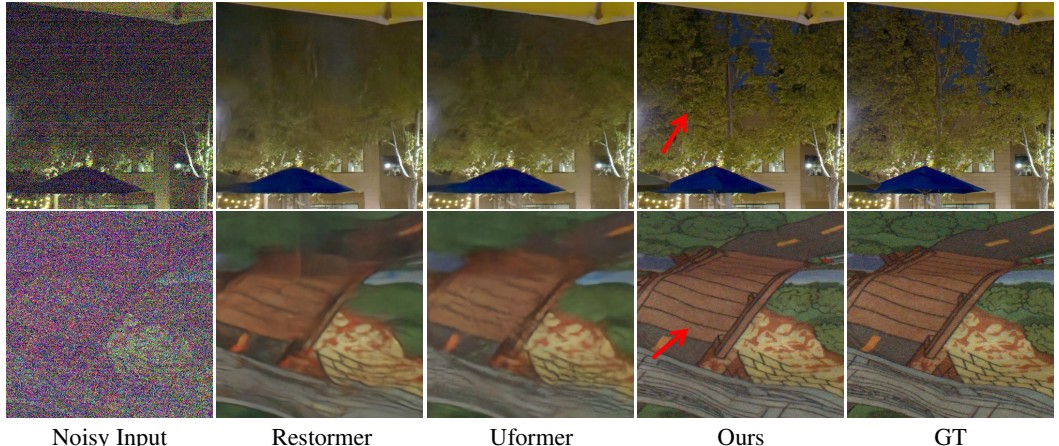

| Noisy Input | Restormer | Uformer | Ours | GT |

Figure 3: Qualitative comparison on indoor and outdoor scenes of SID dataset. (**Zoom in for details**)

using generative models, inferring patches independently often results in noticeable grid artifacts along the boundaries of the splitting patches. Even with a large overlap, the neighboring patches are not consistent in the splitting region.

For high-resolution images, directly adopting the diffusion model on the latent space [39] sacrifices the detailed textures when downsampling the image features to the latent space. Another strategy mask-shift [47] is proposed to alleviate the local incoherence but requires significant computational cost. To address this problem, we propose a simple yet effective inter-step patch-splitting strategy for high-resolution image inference. As shown in Fig. 2, we integrate the patch-splitting technique into diffusion iterations to support high-resolution image generation. Specifically, at each diffusion time step, the input image is split to small overlapped patches. Then, the diffusion model performs denoising on the patches, and the overlapping regions of the results are dropped directly when merging the patches. Since the denoising model focuses on the local receptive field, performing the patch-splitting at the denoising step would not introduce obvious boundary artifacts. Some visualizations can be found in Fig. 6.

## 4 Experiments

### 4.1 Tasks and Datasets

We evaluate our method on three challenging tasks. For extreme low-light denoising, we use the SID Sony dataset [7], which provides noisy-clean image pairs with an exposure factor of 300. It consists of high-resolution images ($2120 \times 1416$) captured in various indoor and outdoor scenes. For deblurring, we follow DvSR [50] to train and test on the GoPro dataset. For JPEG restoration, we train on the ImageNet and follow DDRM [26] to test on selected 1K evaluation images [37].

### 4.2 Experimental Settings and Metrics

To demonstrate our method to be a unified framework, we train the framework under the similar setting without task-specific hyper-parameter tuning. We used the AdamW optimizer with a learning rate of $1 \times 10^{-4}$, and EMA decay rate is 0.9999. In the training, we used the diffusion process with $T = 2000$ steps with the continuous noise level [10]. During the testing, the inference step is reduced to 50 with uniform interpolation. For each task, we provide a regression baseline that adopts the same architecture and training setting for comparison. We train each task for 500k iterations with batch size 32. The training process takes approximately three days to complete when utilizing 8 A100 GPUs.

In this paper, we focus on the perceptual quality of the results. Four perceptual metrics, LPIPS, NIQE, FID, and KID are adopted for evaluation. LPIPS and NIQE are reference-based and non-reference metrics respectively. For NIQE, different implementations produce different results. In this paper, we adopt it from the BasicSR [46] for all methods. For the inception distance FID and KID, we follow the previous paper [33, 50] to extract patches and compute the FID and KID on patch level to obtain stable evaluation results. We also show the distortion metrics, PSNR and SSIM, for reference.

Table 1: The quantitative results on extreme low-light denoising SID dataset.

| Model | Perceptual | | | | Distortion | |
|---|---|---|---|---|---|---|
| | LPIPS↓ | NIQE↓ | FID↓ | KID↓ | PSNR↑ | SSIM↑ |
| SID [7] | 0.361 | 6.58 | 83.44 | 22.47 | 28.88 | 0.780 |
| Restormer [52] | 0.342 | 6.69 | 62.05 | 10.97 | 29.64 | 0.796 |
| NAFNet [8] | 0.362 | 6.67 | 78.30 | 19.06 | 29.14 | 0.778 |
| Uformer [49] | 0.338 | 6.45 | 74.44 | 17.01 | 29.27 | 0.793 |
| Regression | 0.351 | 6.78 | 75.79 | 17.88 | 29.13 | 0.778 |
| Ours | 0.222 | 6.43 | 55.07 | 6.79 | 28.78 | 0.744 |

Table 2: Quantitative results on GoPro dataset.

| Model | Perceptual | | | | Distortion | |
|---|---|---|---|---|---|---|
| | LPIPS↓ | NIQE↓ | FID↓ | KID↓ | PSNR↑ | SSIM↑ |
| Uformer [49] | 0.087 | 5.18 | 21.60 | 9.96 | 33.05 | 0.962 |
| HINet [9] | 0.088 | 5.08 | 17.91 | 8.15 | 32.77 | 0.960 |
| MPRNet [54] | 0.089 | 5.16 | 20.18 | 9.10 | 32.66 | 0.959 |
| MIMO-UNet+ [11] | 0.091 | - | 18.05 | 8.17 | 32.45 | 0.957 |
| SAPHNet [43] | 0.101 | - | 19.06 | 8.48 | 31.89 | 0.953 |
| DeblurGANv2 [27] | 0.117 | 4.82 | 13.40 | 4.41 | 29.08 | 0.918 |
| DvSR [50] | 0.084 | 4.80 | 12.20 | 4.33 | 30.66 | 0.941 |
| Regression | 0.097 | 5.17 | 17.98 | 7.16 | 31.50 | 0.948 |
| Ours | 0.080 | 4.63 | 8.06 | 2.42 | 30.27 | 0.935 |

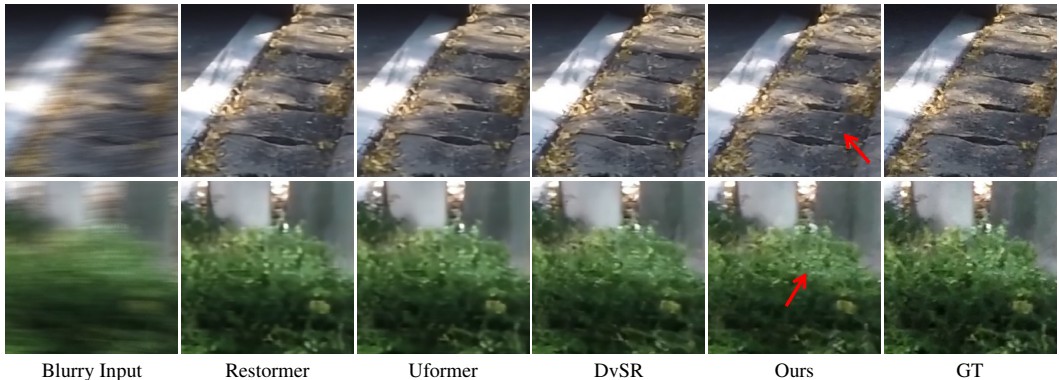

| Blurry Input | Restormer | Uformer | DvSR | Ours | GT |

Figure 4: Visualization of deblurring results on GoPro dataset.

## 4.3 Comparisons on Extreme Low-light Denoising

To evaluate our method's performance on extreme low-light denoising, we compare it with the official implementations of transformer-based and convolution-based backbones [52, 8, 49] on SID dataset [7]. As the SID dataset consists of high-resolution images, we utilize our inter-step patch-splitting strategy to perform full-resolution image inference without grid artifacts. The quantitative results, presented in Tab. 1, demonstrate that our method achieves the best performance across four perceptual metrics. As shown in Fig. 3, while the regression models generate higher distortion results, their perceptual quality is quite poor. All the previous methods tend to produce very blurry results. In contrast, our method produces clear images with fine details, vivid colors, and well-preserved structures, demonstrating its superior perceptual quality.

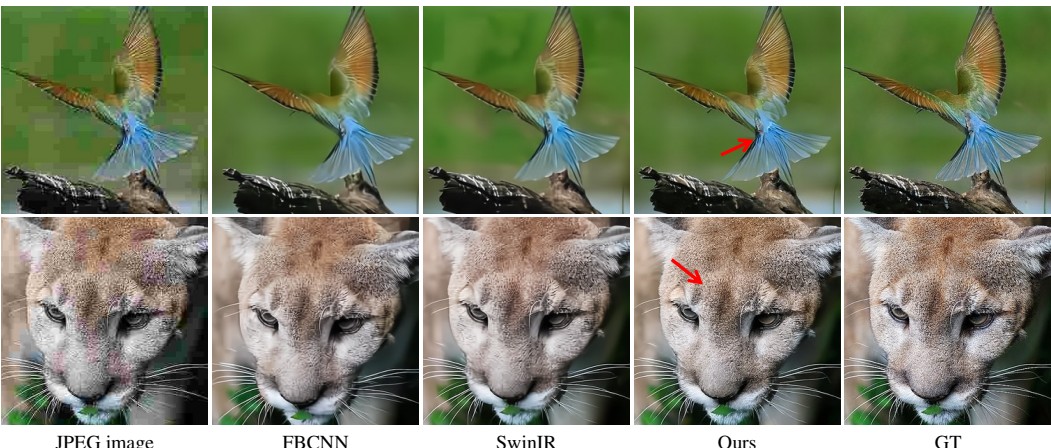

| JPEG image | FBCNN | SwinIR | Ours | GT |

Figure 5: Qualitative comparison on the JPEG restoration.

## 4.4 Comparisons on Image Deblurring

We also evaluate our model on GoPro dataset [35] and compare it with the popular deblurring methods. Since the diffusion-based method DvSR [50] has not released the code, we implemented it based on an open-sourced project. The origin training setting of DvSR requires 32s TPUs, which is hard to reproduce. We train DvSR under the same lightweight training setting as our method. The reproduced perceptual quality has similar conclusions as the original paper. The quantitative results are shown in Tab. 2. Our method achieves the best perceptual quality over all perceptual metrics. Fig. 4 shows the visualization of the deblurring results.

## 4.5 Comparisons on JPEG restoration

For the JPEG restoration task, we perform a comparison between our method and several baseline models, including regression models (FBCNN [23] and SwinIR [31]) and the diffusion-based method DDR [26].

We follow previous works [26] to conduct the evaluation on restoring $256 \times 256$ images with a Quality Factor of 10. To ensure a fair comparison, we utilize the official implementations of these baseline methods. The quantitative and qualitative results are consistent with previous tasks. As shown in Tab. 3, our method shows significant improvements in terms of perceptual quality. The visualization results in Fig. 5 further illustrate this conclusion. Unlike previous work [26], which can only work on inputs of a fixed size. Our method supports arbitrary resolutions by adopting the inter-step patch-splitting strategy. Some high-resolution results are provided in Fig. 5 .

Table 3: JPEG restoration results on ImageNet.

| Model | Perceptual | | | | Distortion | |
|---|---|---|---|---|---|---|
| | LPIPS↓ | NIQE↓ | FID↓ | KID↓ | PSNR↑ | SSIM↑ |
| SwinIR [31] | 0.205 | 8.95 | 65.81 | 22.9 | 28.00 | 0.883 |
| QGAC [16] | 0.260 | - | - | - | 28.01 | 0.800 |
| FBCNN [23] | 0.211 | 8.70 | 68.29 | 24.13 | 27.81 | 0.878 |
| DDRM [26] | 0.260 | 8.34 | 74.45 | 24.87 | 27.68 | 0.780 |
| Regression | 0.217 | 9.68 | 74.38 | 27.47 | 27.91 | 0.807 |
| Ours | 0.139 | 6.92 | 38.74 | 6.30 | 27.75 | 0.808 |

## 4.6 The Effectiveness of Inter-step Patch-splitting Strategy

In Fig. 6, we provide a visual comparison between the common patch-splitting strategy and our proposed inter-step patch-splitting strategy. As we can see, adopting the common practice always produces severe artifacts around the patch boundaries, and the neighboring patches cannot produce consistent content. Further increasing the amount of overlapped patch padding does not alleviate this issue. In contrast, our inter-step patch-splitting strategy produces more consistent and visually coherent output.

Table 4: The ablation studies on extreme low-light denoising dataset.

| Model | LPIPS↓ | MACs↓ |
|---|---|---|
| w/ → w/o LayerNorm | 0.227 | 71.7 |
| LayerNorm → GroupNorm | 0.240 | 71.7 |
| Swish → ReLU | 0.235 | 71.7 |
| w/o Spatial guidance | 0.240 | **70.9** |
| Internal feature guidance | 0.247 | 72.1 |
| Degraded image guidance | 0.253 | 71.7 |
| Addition | 0.236 | 71.5 |
| Concatenation | 0.241 | 112.7 |
| AdaIN [21] | 0.240 | 71.5 |
| DvSR [50] | 0.244 | 73.7 |
| Ours | **0.222** | 71.7 |

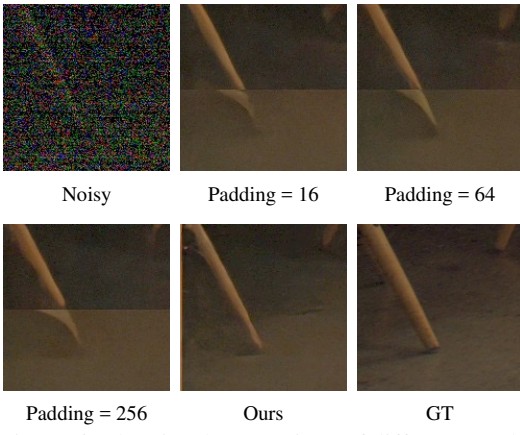

Noisy    Padding = 16    Padding = 64

Padding = 256    Ours    GT

Figure 6: The visual comparison of different patch-based splitting strategy.

### 4.7 Ablation Studies

In this section, we conduct ablation studies to analyze the impact of different design choices in our conditional diffusion framework. We use the LPIPS metric to compare the performance of different configurations, and we also report the corresponding computational costs.

**The design of the basic module.**    We first validate the design of the basic module. As shown in the second row block of Tab. 4, removing LayerNorm (denoted as "w/$\rightarrow$w/o LayerNorm") or replacing the Swish activation functions with ReLU ("Swish$\rightarrow$ReLU") leads to decreased performances. We also explored using GroupNorm ("LayerNorm$\rightarrow$GroupNorm"), as in S3 [42]. However, we observed that GroupNorm tends to introduce color distortion artifacts in the restored images. This might be caused by the grouping of channels, which might not capture the consistent relationships across channels.

**The importance of spatial guidance.**    We further investigated the impact of removing the spatial guidance branch from our framework ("w/o Spatial guidance") and keeping all other settings the same. The results, shown in the third row block of Tab. 4, indicate that removing the spatial guidance branch decreases the LPIPS score to 0.230, but it does not significantly reduce the MACs. We also explored alternative choices for the spatial guidance, including using the internal feature map ("Internal feature guidance") or the degraded image ("Degraded image guidance") itself as the spatial guidance. However, these alternatives do not yield satisfactory results. Leveraging the output of the initial predictor as the spatial guidance is more effective. The initial predictor captures the low-frequency structural information of the final output, which is more stable and deterministic compared to using the internal feature map or the degraded image. By injecting the initial guidance into all the diffusion model blocks, we ensure consistent spatial guidance throughout the restoration process.

**Other integration choices.**    We also compare the previous condition integration designs with our method. As shown in the fourth row block of Tab. 4, directly adding ("Addition") or concatenating ("Concatenation") the guidance feature map, or using AdaIN ("AdaIN") [21] to generate conditional normalization coefficients as the input to the diffusion block cannot effectively capture and utilize the conditional information for guiding the generation process. The alternative approach fails to fully leverage the potential of the diffusion model and results in suboptimal restoration quality. In contrast, our method, which incorporates the guidance feature map through the Adaptive Kernel Guidance Module, provides a more effective and adaptive integration of the conditional information. By dynamically generating adaptive kernels based on the fused multi-source conditional information, our method achieves spatially adaptive conditioning within each block of the diffusion model.

## 5    Conclusion

In this paper, we introduce a unified conditional framework for image restoration tasks. Our framework incorporates an initial predictor to generate the initial guidance and facilitates the integration of multiple sources of conditional information into each block of the diffusion model. We propose a basic module and an Adaptive Kernel Guidance Block to effectively combine the spatial guidance and auxiliary scalar information, enhancing the generative capacity of the diffusion model. To handle high-resolution images, we propose an inter-step patch-splitting strategy that ensures consistent and artifact-free high-resolution results. Extensive experiments on extreme low-light denoising, image deblurring, and JPEG restoration demonstrate the effectiveness and generalization capability of our proposed framework. The experimental results highlight the superior perceptual quality achieved by our method when compared to strong regression baselines and recent diffusion-based models.

## 6    Limitation and Future Direction

While our method achieves superior quantitative and qualitative results on three challenging tasks, there are still several problems that can be further explored. In this work, we simply use a uniform noise schedule with fewer time steps to speed up the sampling process of the diffusion model. This can be further accelerated by adopting recent fast sampling methods. Besides, since the generation does not explicitly consider semantic guidance, it may generate some unnatural textures (*e.g.*, incorrect characters), which also has been mentioned in previous works [42]. How to control the generation ability of the diffusion model is also a very interesting direction.

# 7 Acknowlegement

This project is funded in part by National Key R&D Program of China Project 2022ZD0161100, by the Centre for Perceptual and Interactive Intelligence (CPII) Ltd under the Innovation and Technology Commission (ITC)'s InnoHK, by General Research Fund of Hong Kong RGC Project 14204021. Hongsheng Li is a PI of CPII under the InnoHK.

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
