# Supplementary Material of A Unified Conditional Framework for Diffusion-based Image Restoration

Yi Zhang [1]   Xiaoyu Shi [1]   Dasong Li [1]   Xiaogang Wang [1]   Jian Wang [2] *   Hongsheng Li [1] *

[1] The Chinese University of Hong Kong    [2] Snap Research

zhangyi@link.cuhk.edu.hk,   jwang4@snapchat.com,   hsli@ee.cuhk.edu.hk

https://zhangyi-3.github.io/project/UCDIR

## 1   Implementation Details

For all tasks, we adopt a UNet architecture similar to the one described in DvSR [4]. The input feature map is expanded to 64 channels. There are five stages in both the encoder and decoder, and each stage contains two diffusion model blocks. Between each encoder stage, the input resolution is downsampled by a convolution layer with stride 2 and the channels are expanded by a factor of 2. On the other hand, in each decoder stage, the feature map resolution and the channels are reversed by the Nearest upsampling and a convolution layer separately.

During training, we use a linear noise schedule with a total of $T = 2000$ steps. The noise level is sampled uniformly from the range $[1 \times 10^{-6}, 1 \times 10^{-2}]$. For evaluation, we simply use a shorter noise schedule with $T = 50$ steps and a noise range of $[1 \times 10^{-6}, 4 \times 10^{1}]$. This allows for faster inference during evaluation.

## 2   Additional Results

Since the DDRM [2] can only work on a fixed input size of $256 \times 256$, we are unable to present the high-resolution results in the main text. Here, we provide the comparisons on the low-resolution images in Fig. 1.

We provide more visualization results in Fig. 2, Fig. 3, Fig. 4, and Fig. 5. As mentioned in the limitation section of the main text, our method can generate realistic textures in most regions. However, it may restore incorrect small characters as shown in Fig. 2, which is highly ill-posed. This has also been observed in previous works such as S3 [3]. Exploring better control mechanisms for the generation process, such as disabling generation around specific regions, could be an interesting direction for future research. We also evaluate the generalization of our method on the HIDE dataset in Tab. 1. Compared with the Uformer, it shows consistent improvements in perceptual quality.

Table 1: Motion deblurring results on HIDE dataset.

| Model | Perceptual | | | | Distortion | |
|---|---|---|---|---|---|---|
| | LPIPS↓ | NIQE↓ | FID↓ | KID↓ | PSNR↑ | SSIM↑ |
| Uformer | 0.114 | 5.46 | 7.56 | 1.08 | 30.89 | 0.940 |
| Ours | 0.095 | 4.59 | 6.92 | 0.26 | 28.69 | 0.884 |

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

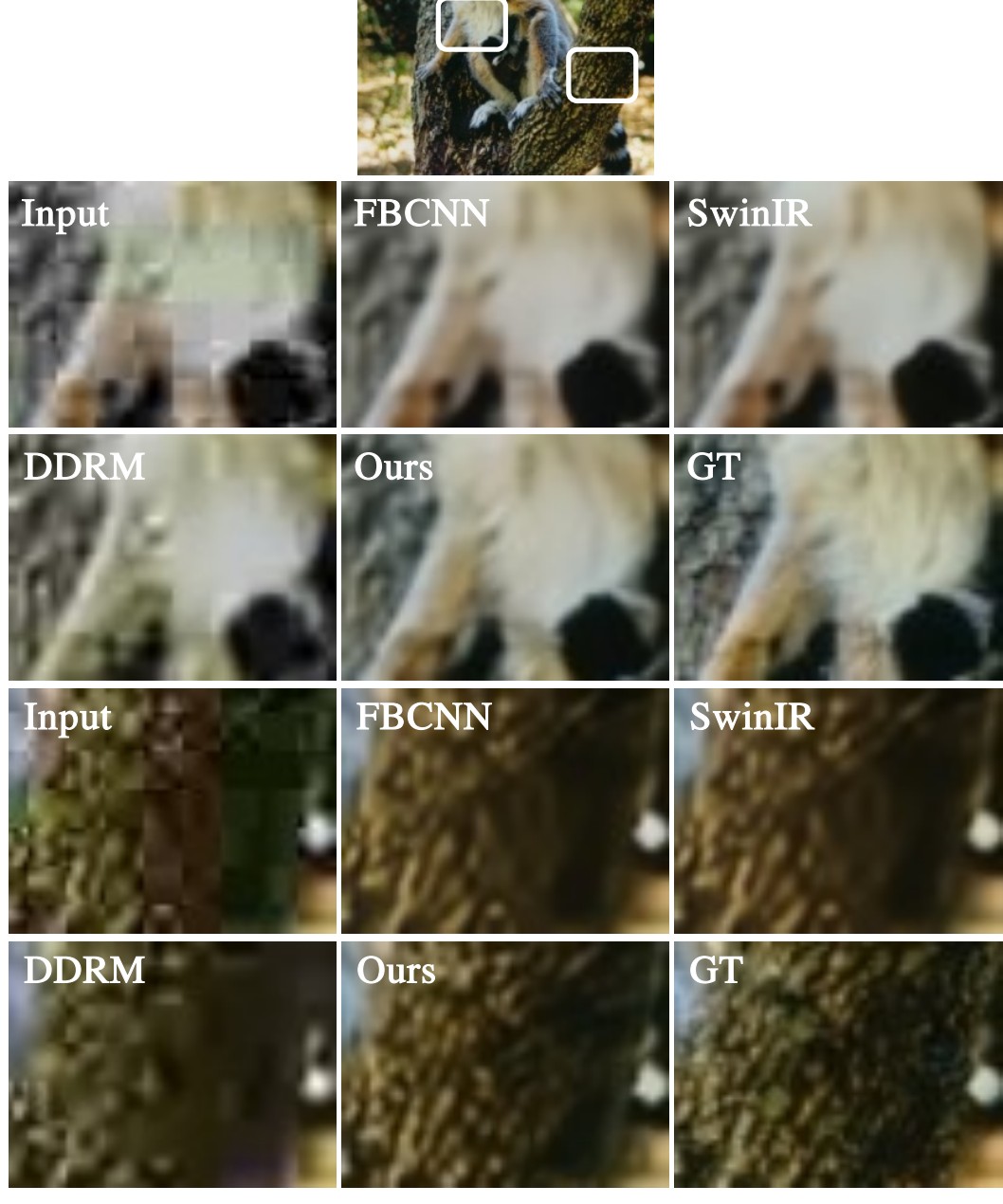

Figure 1: Visualization comparison with DDRM [2] on JPEG restoration.

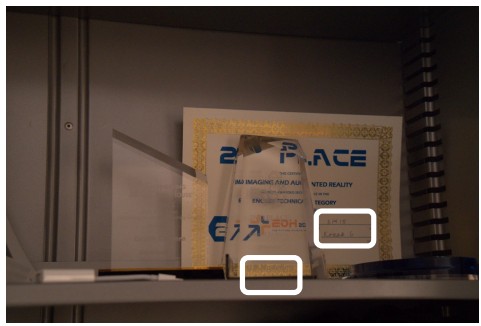

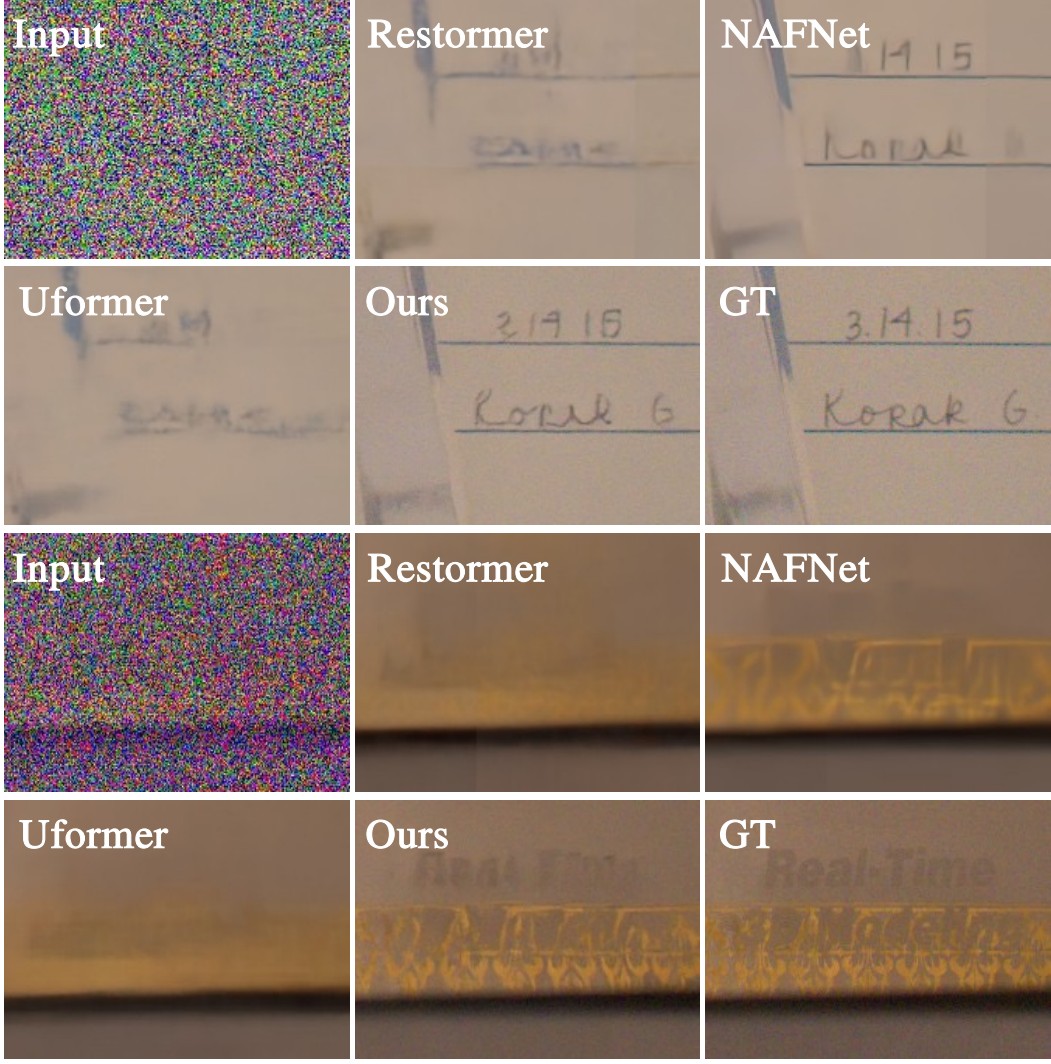

Figure 2: Additional extreme low-light denoising results on the SID [1] dataset.

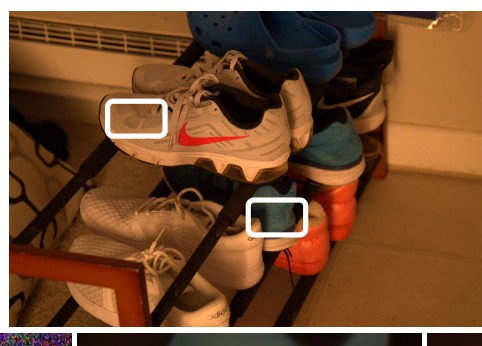

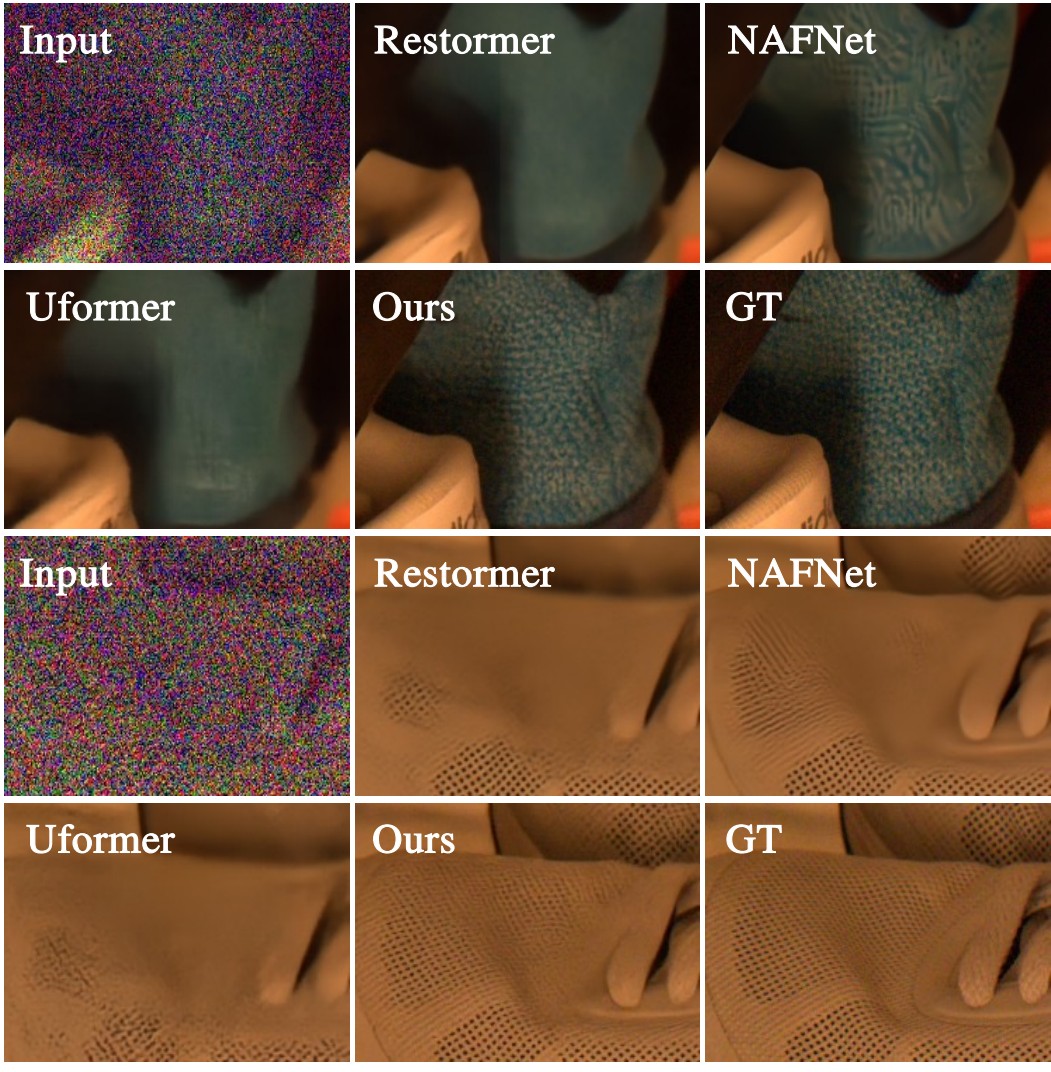

Figure 3: Additional extreme low-light denoising results on the SID [1] dataset.

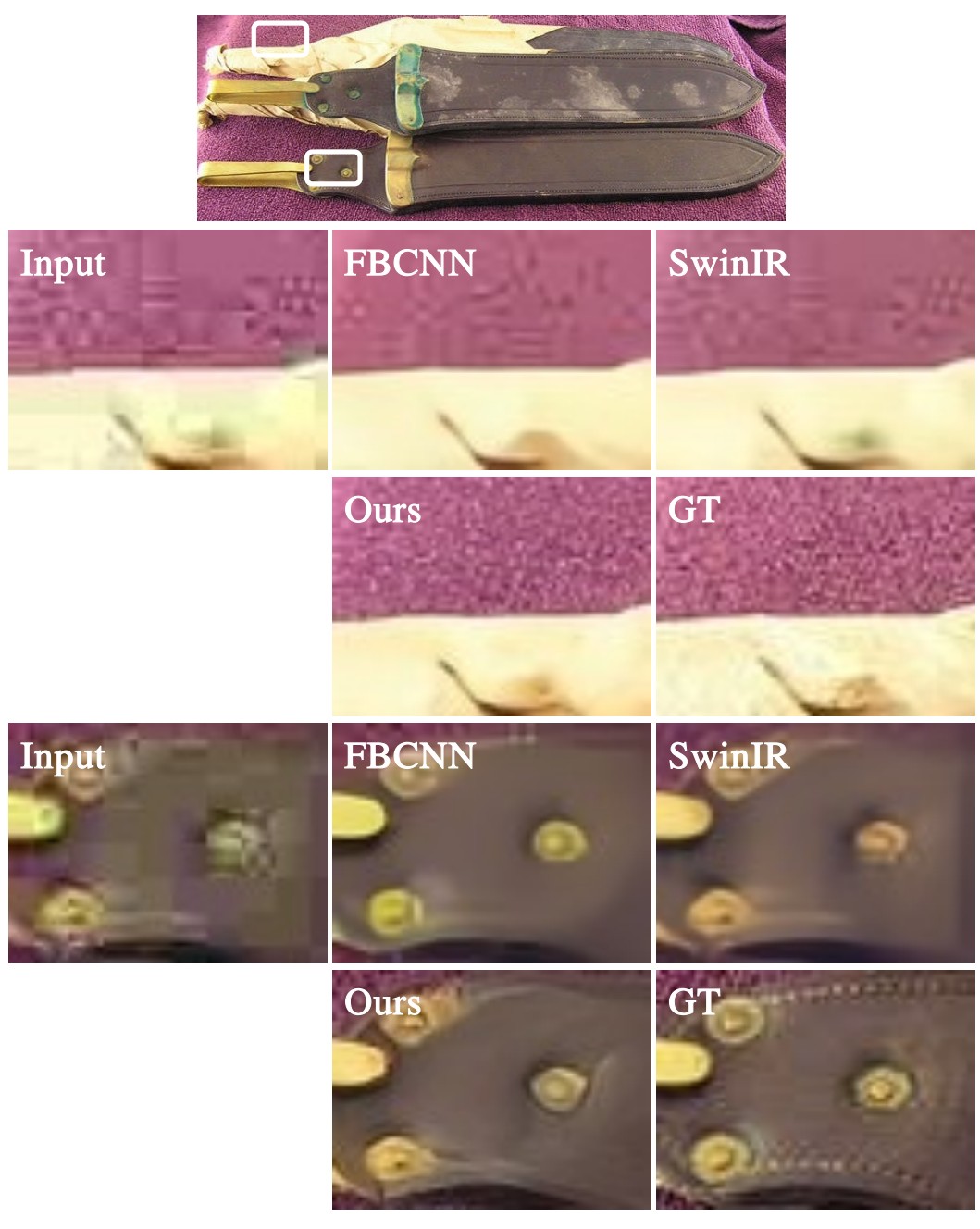

Figure 4: Additional JPEG restoration results on the ImageNet validation dataset.

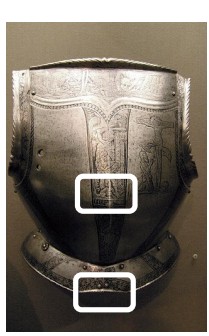

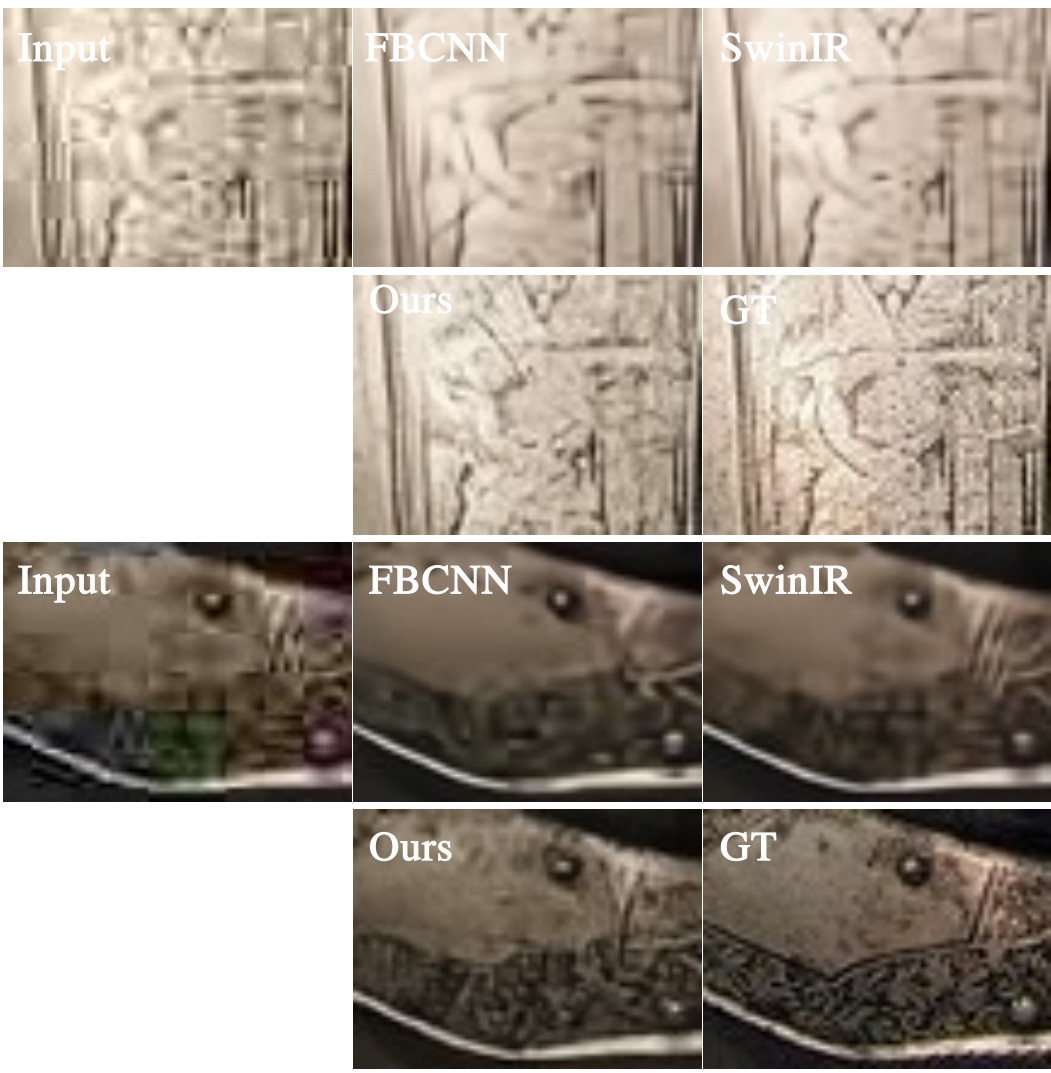

Figure 5: Additional JPEG restoration results on the ImageNet validation dataset.

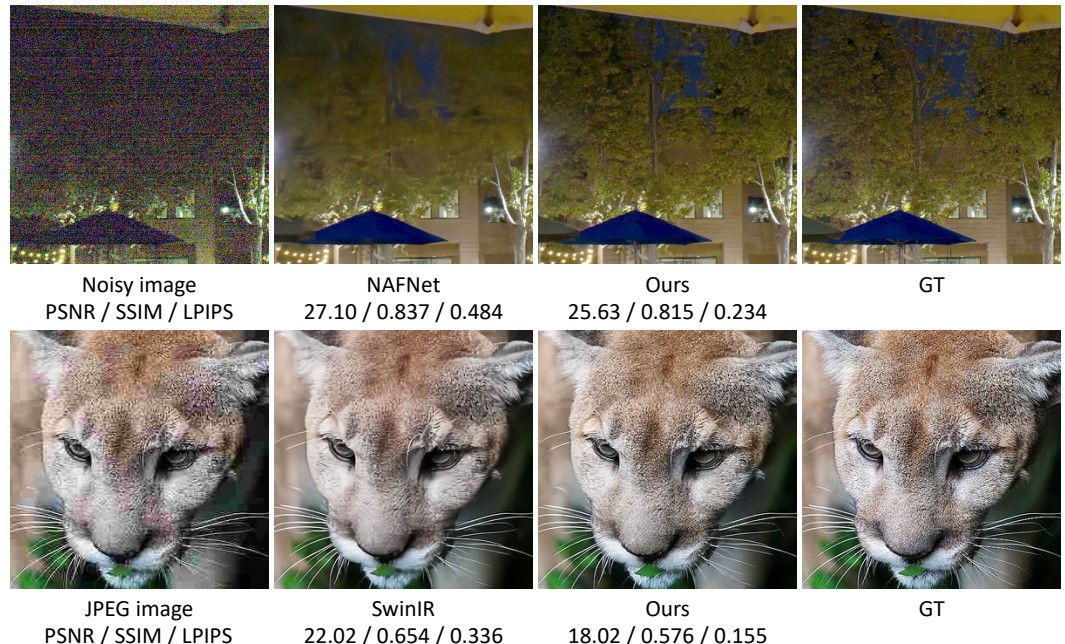

| Noisy image | NAFNet | Ours | GT |
| PSNR / SSIM / LPIPS | 27.10 / 0.837 / 0.484 | 25.63 / 0.815 / 0.234 | |

| JPEG image | SwinIR | Ours | GT |
| PSNR / SSIM / LPIPS | 22.02 / 0.654 / 0.336 | 18.02 / 0.576 / 0.155 | |

Figure 6: We compare the PSNR-oriented methods and our method. They produce higher PSNR/SSIM but tend to be blurry. Our method fits the potential distribution for the degraded input and shows significant improvements on the perception quality. (**Zoom in for details.**)

[2] Bahjat Kawar, Michael Elad, Stefano Ermon, and Jiaming Song. Denoising diffusion restoration models. In *NeurIPS*, 2022.

[3] Chitwan Saharia, Jonathan Ho, William Chan, Tim Salimans, David J. Fleet, and Mohammad Norouzi. Image super-resolution via iterative refinement. *IEEE Trans. Pattern Anal. Mach. Intell.*, 45(4):4713–4726, 2023.

[4] Jay Whang, Mauricio Delbracio, Hossein Talebi, Chitwan Saharia, Alexandros G. Dimakis, and Peyman Milanfar. Deblurring via stochastic refinement. In *CVPR*, 2022.