# OpenReview forum: "A Unified Conditional Framework for Diffusion-based Image Restoration"
_NeurIPS.cc/2023/Conference — NeurIPS 2023 poster_

### Official Review · Reviewer_37QZ · 2023-07-03

**Soundness:** 3 good
**Presentation:** 2 fair
**Contribution:** 3 good
**Rating:** 5
**Confidence:** 3

**Summary:**

The paper proposes a new framework for diffusion-based image restoration. The underlying architecture is based on that of "DvSR" [49], in which there is a deterministic predictor that does an initial restoration, used in conjunction with a probabilistic diffusion denoising network that applies the diffusion reverse process to estimate the additive residual to the initial prediction, that can be combined to get the final restoration.

The novelty is the change in the residual denoiser network (itself based on SR3 [42]). Instead of using spatially invariant convolutions that are fixed after learning, it employs the BPN [50] idea of having basis convolutional kernels that can be linearly combined at different spatial positions, with combination weights that are themselves estimated from a fusion of the initial deterministic restoration, and other auxiliary guidance (diffusion timestep and degradation type were mentioned).

In addition, a new inter-step patch-splitting procedure that is integrated with the diffusion denoising process is also used for patch-wise high resolution restoration, to overcome problems with artifacts at patch boundaries.

Experiments were conducted on the SID dataset [7] for low-light denoising, on the GoPro dataset [37] for motion deblurring, and on ImageNet JPEG compression artifact restoration.

**Strengths:**

Generally there is decent novelty. Although the main ideas are taken from existing methods, the combination of these ideas are relatively new and not immediately obvious. The use of basis kernels from [50] for creating spatially-variant convolutional kernels for the diffusion denoiser is quite interesting.

The inter-step patch splitting, although simple, appears to be new and effective, with Fig. 6 quite convincing.

Experimental results substantially exceed current SOTA on perceptual metrics (LPIPS, NIQE, FID, KID), though more mediocre on distortion metrics (PSNR, SSIM). It is noted though that the DvSR network was retrained by the authors to match the same training hyperparameters that are more lightweight, due to inaccessible computational requirements of DvSR; otherwise the published results in [49] are better.

**Weaknesses:**

On experiments:
- Since the method is essentially derived from DvSR, and the authors are retraining it based on their hyperparameters, it is important to conduct experiments with DvSR on the other low-light denoising and JPEG restoration tasks as well.
- Generalization tests from GoPro to the HIDE and RealBlur datasets have been done in various other baseline papers. The authors should have also tested on these.
- The NIQE numbers don't match the ones in [49], though the other perceptual metrics do. If they are different, the authors should properly justify in the paper why this is the case.

Multiple parts of the paper are unclear, as details have been left out.
- How is the auxiliary scalar information represented? For example, how are timesteps encoded? Details should at least be in the supplementary material.
- What exactly is happening in the inter-step patch-splitting procedure? Figure 2 is very vague. Although the results look good, and a reader can perhaps roughly guess what is happening, there are no details given.
- The descriptions in Section 4.7 Ablation Studies are very vague and need to be explained better.
    - What happens in "w/o spatial guidance"? How does time and other scalar information get injected (or not)?
    - What are "internal feature guidance", and "degraded image guidance"? These need clearer explanation.

The paper is littered with spelling errors. This can simply be resolved by running a spell-checker; that it is not done suggests poor neglect on the part of the authors. Some instances:
- L120: "Restoraion" => "Restoration"
- L210: "receipt" => "receptive"
- L257: "JEPG" => "JPEG"

**Questions:**

Please answer the questions raised in the Weaknesses section.

The baseline methods are for the different restoration tasks are barely overlapping. How is the decision for the choice of baselines for each dataset? For example, Uformer is tested on both SID and GoPro, but not Restormer. Why aren't they be tested across the different tasks / datasets?

While it is generally accepted that spatially variant kernels can be useful in many situations, I think it is worth a discussion when it comes to denoising residuals. Intuitively we may expect the residuals have a more uniform spatial distribution, so kernels don't have to be spatially varying. This is unlike convolutions on normal images with strong spatially dependent image structures. But it seems the results suggest otherwise. Can the authors provide a better understanding on this?


**Limitations:**

The technical limitations given in Section 6 are quite reasonable.

However, the authors should also include some discussion about potential negative societal impact, since image restoration can also lead to hallucination of details that were not present in the original images.

---

> ### Author Rebuttal · Authors · 2023-08-09
>
> > It is suggested to conduct experiments with DvSR on the low-light denoising and JPEG restoration tasks
>
> Thanks for the advice. We will try the DvSR [4] on low-light denoising and JPEG restoration tasks. In the Table 1 and Figure 1 of the newly attached PDF, we show the latest LPIPS and the visualization of DvSR respectively on the extreme low-light denoising.
>
> > Generalization tests on the deblurring dataset (HIDE and RealBlur)
>
> We will conduct the generalization tests. Some preliminary results on HIDE dataset are show in Table 2 of the newly attached PDF. Our method consistently outperforms the Uformer.
>
> > Inconsistency of NIQE numbers with existing papers
>
> We also noticed this phenomenon in the experiments. Various implementations of NIQE [1-3] report different numerical values but maintain the same rank order. To ensure a fair comparison, we adopted the NIQE implementation from basicSR [2] for all methods in our experiments. We will revise the paper to make it clear.
>
> > Unclear parts and typos of the paper.
>
> We thank the reviewer for pointing out the unclear parts. We will revise them as follows:
> - We follow the SR3 to employ positional encoding and a stack of two linear layers with a Swish activation function to process all auxiliary scalars.
> - To enhance clarity, we will include a figure and provide a more detailed explanation to present the inter-step patch-splitting procedure.
> - For the 'w/o spatial guidance' variant, we exclude the spatial guidance part of the AKGM module while retaining other scalar information, which is injected by adding it to the feature map.
> - For each AKGM block, the spatial guidance is generated using the output of the initial predictor. The 'internal feature guidance' employs the intermediate feature map within each block to generate spatial guidance. Additionally, the 'degraded image guidance' uses the input degraded image to generate spatial guidance.
>
> > Questions:
>
> - How is the decision for the choice of baselines for each dataset?
>
> We decided to adopt existing the state-of-the-art methods as the backbones for each task, which might be of different architectures.
>
> It is a good idea to add Restormer on GoPro dataset, which would be more consistent between different tasks. Since the Uformer shows better performance than Restormer on GoPro dataset, we select Uformer as the representative method at that time. For the SID dataset, they do not show the results on it, so we adopt all the strong backbones for comparisons.
>
> - The understanding of adopting spatially variant kernels on the residual domain.
>
> Although the residuals may visually appear more "uniform" when visualized, their inherent properties persist. Texture areas remain challenging to learn, making the adoption of spatially adaptive kernels more effective in handling the task. Additionally, considering that the residuals contain high-frequency information while the initial predictor output includes low-frequency components, employing spatially adaptive kernels becomes a natural choice to address the high-frequency details adaptively.
>
> [1] https://github.com/chaofengc/IQA-PyTorch
>
> [2] https://github.com/xinntao/EDVR/blob/master/basicsr/metrics/niqe.py
>
> [3] https://github.com/aizvorski/video-quality/blob/master/niqe.py
>
> [4] Whang et. al., Deblurring via stochastic refinement. CVPR, 2022.

---

> > ### Comment · Reviewer_37QZ · 2023-08-11
> >
> > I would like to thank the authors for their additional experiments and further clarifications. If the paper is accepted, please provide more clarification details and more extensive experimental results, beyond the scope of a rebuttal.

---

### Official Review · Reviewer_KQ6K · 2023-07-04

**Soundness:** 3 good
**Presentation:** 3 good
**Contribution:** 2 fair
**Rating:** 3
**Confidence:** 5

**Summary:**

This paper proposes a unified framework for diffusion-based framework. In the proposed framework, an initial predictor is used to produce a rough restoration of the input degraded image. The roughly restored image is then used as condition to the diffusion model using a Conditional Integration Module. Experiments show that the proposed approach achieves promising results on low-light denoising, debarring, and JPEG restoration.

**Strengths:**

1. The proposed method achieves promising performance on three different restoration tasks. The proposed framework is versatile in the sense that it can handle multiple types of degradations.

2. This paper is easy to follow.

**Weaknesses:**

1. The technical novelty is limited. In particular, the residual prediction formulation was proposed in [49]. The inter-step patch-splitting strategy has been discussed in [*1].

2. Many of the proposed components do not lead to significant performance gain. Specifically, in Table 4, the worst model obtains a LPIPS of 0.253, which is still significant better than existing works in Table 1. The performance gain from the proposed components are insignificant.

3. Given the insignificant performance gain mentioned above, the performance gain could come from the increase of model complexity. Therefore, it is advised to provide additional analysis on the model to demonstrate the superiority of the proposed method.


[*1] Álvaro Barbero Jiménez, Mixture of Diffusers for scene composition and high resolution image generation, 2023

**Questions:**

While the proposed method achieves promising performance on multiple restoration tasks, the novelty is limited and the source of performance gain is unclear. Please refer to the weakness section for the questions.

**Limitations:**

Limitations are appropriately discussed.

---

> ### Author Rebuttal · Authors · 2023-08-09
>
> > The technical novelty is limited (residual prediction formulation and the inter-step patch-splitting strategy.).
>
> As described in L67-L81, we actually did not claim the residual prediction as one of our contributions. The residual formulation is a common practice and has been adopted in many previous methods in low-level vision. In this paper, our contributions are a unified conditional framework and an Adaptive Kernel Guidance Block (AKGB) to incorporate the conditional information into diffusion models.
>
> The inter-step patch-splitting strategy shares the similar spirit with the unpublished paper [1]. The key difference is that [1] needs to fuse the overlapped regions by the Gaussian weights since it adopts several diffusion models on high-level vision tasks, which need to understand the large receptive field sematic information. Our method is designed for low-level vison tasks, which mostly focus on local textures.  Therefore, we can perform the padded patch inference without considering fusing in each diffusion step.
>
> > On the SID dataset, the proposed method outperforms existing regression works significantly. The performance gain from the proposed components is insignificant.
>
> When facing severe degradation or highly ill-posed tasks (such as extreme low-light denoising), generative methods tend to outperform regression methods due to the availability of multiple potential candidates for each input. Generative methods can generate more diverse and realistic appearance, , while regression methods tend to predict averaged results, leading to blurry outcomes. Consequently, on the SID dataset, a substantial improvement in terms of LPIPS is observed. Conversely, for the GoPro dataset (Table 2 in our paper), where the problem is less ill-posed, such the performance gain is smaller.
>
> Please note that, unlike PSNR, LPIPS is not scaled by the log function, resulting in smaller quantitative improvements. However, even small LPIPS improvements correspond to a significant gain. For instance, in Table 2 of our paper, our method improves LPIPS by 0.007 compared to Uformer, yet the corresponding FID improvement exceeds 12.5, highlighting the substantial qualitative gains achieved.
>
> > Additional analysis on the model complexity
>
> We compare the model complexity with the DvSR [3] in the ablation studies. As indicated in Table 1 of the newly attached PDF, our approach exhibits comparable computational cost to DvSR while achieving significantly better perceptual metrics. The corresponding visual improvements are presented in Figure 1.
>
> [1] Mixture of Diff users for scene composition and high-resolution image generation
>
> [2] EnhanceNet: Single Image Super-Resolution Through Automated Texture Synthesis

---

> > ### Comment · Reviewer_KQ6K · 2023-08-12
> >
> > Thanks for the authors' response. I have the following additional questions:
> >
> > 1. I do not understand the sentence `Therefore, we can perform the padded patch inference without considering fusing in each diffusion step.` in the response. From Fig.2, I believe that fusion is performed at each step?
> >
> > 2. Regarding the performance gain:
> >
> >     **a.** What I mean is even the worst model in Table 4 is already significantly better than the models in Table 1 (i.e., LPIPS 0.253 of `Degraded Image Guidance` vs 0.338 of `Uformer`). The performance gain brought by the proposed components is 0.253-0.222=0.031 vs the improvements from baseline over previous work 0.338-0.253=0.085.
> >
> >     **b.** When we further compare `Regression` vs `Uformer` in Table 1 and Table 2, we see that the proposed architecture underperforms previous works. This could mean that the proposed guidance is ineffective. Please correct me if I am wrong.
> >
> > 3. Speaking of universal framework, the submission seems to be missing related works on degradation modeling for super-resolution. In particular, DASR [*1] encodes the input into an embedding. Conceptually the proposed method shares some similarities. It is advised to discuss and compare with methods in the corresponding domain.
> >
> > [*1] Liang et al., Efficient and Degradation-Adaptive Network for Real-World Image Super-Resolution, ECCV, 2022

---

> > > ### Author Response · Authors · 2023-08-18
> > > **Response to the additional questions from Reviewer KQ6K**
> > >
> > > Thanks for your additional questions.
> > >
> > > > 1. The patches fusion at each diffusion step.
> > >
> > > Our proposed inter-step splitting strategy does not involve weighted fusion in the overlapped regions of patch. Instead, for each overlapped patch, only the centered and non-overlapped regions would be cropped and kept in the results.
> > >
> > > > 2.a The improvement from the baseline is larger than the improvement from the proposed components
> > >
> > > The reason for the large improvements of the baseline model is that the task (extreme low-light denoising) is highly ill-posed . Regression methods tend to produce more blurry images due to the availability of multiple candidate results . This leads to regression method having higher PSNR and lower LPIPS (see Figure 2 of our newly attached PDF). As a result, the baseline model looks to improve a lot. When the task is less ill-posed (Table 2 GoPro dataset), the corresponding improvements would not be similarly large.
> > >
> > > But this does not indicate the improvement of the proposed modules is insignificant.
> > > 1. Small LPIPS improvements still correspond to a significant gain. For instance, in Table 2 of our paper, our method improves LPIPS by 0.007 compared to Uformer, yet the corresponding FID improvement exceeds 12.5, highlighting the substantial qualitative gains achieved.
> > >
> > > 2. We compare with DvSR [2] in the newly attached PDF in Table 1 and Figure 1. Both the quantitative and the qualitative results demonstrate significant improvements of our method.
> > >
> > > > 2.b The proposed architecture underperforms previous works like Uformer.
> > >
> > > A direct comparison between the Uformer and regression models is not feasible.
> > > The proposed components are designed for diffusion models to integrate the conditional information effectively instead of outperforming previous methods in regression tasks. Here, we only want to provide a regression result and show the difference between the regression and generation models under similar settings. As a result, the training settings as well as the distinct characteristics of their  architectures are quite different with the Uformer.
> > > It is out of our paper’s scope to decide its effectiveness on regression tasks.
> > >
> > > > 3. Discussion with degradation modeling methods
> > >
> > > DASR is designed to learn the degradation representations for the real-world super-resolution task. While a direct comparison between our method and DASR [1] is not applicable, it is very interesting to discuss degradation modeling from the universal framework perspective in our paper.
> > >
> > > Please let us know if the above responses have further clarified our paper.
> > >
> > > [1] Liang et al., Efficient and Degradation-Adaptive Network for Real-World Image Super-Resolution, ECCV, 2022.
> > >
> > > [2] Whang et. al., Deblurring via stochastic refinement. CVPR, 2022.

---

> > > > ### Comment · Reviewer_KQ6K · 2023-08-19
> > > >
> > > > Thank you for the prompt response. Below are some follow-up questions.
> > > >
> > > > `2.a.1` Since regression-based models tend to produce blurry results, I think comparing diffusion model with Uformer is not able to demonstrate the performance gain. In particular, the performance gain is smaller when comparing DvSR and the proposed model. It is recommended to provide also the FID for the variants on Table 4 as well as the Table 1 in the new attachment to provide a more thorough understanding. Also, while the proposed method is perceptual-oriented, most methods in comparison are PSNR-oriented. It cannot effectively show to advantage of the proposed model.
> > > >
> > > > `2.a.2` As shown in the Table, most variants are better than DvSR. In particular, even without guidance, the LPIPS is still better than DvSR. It seems to show that the effectiveness of the approaches are limited?
> > > >
> > > > `2.b` Could you elaborate on this? I am curious why the proposed components work only for diffusion model. In particular, the guidance shouldn't be limited to diffusion-based models. Since the training objectives of diffusion models are different from that for PSNR-based models, I think the current experiments are unable to demonstrate the effectiveness of the components. So I think a fair comparison using regression should not be out of the scope of this paper.
> > > >
> > > > `3` Conceptually DASR is similar to the proposed approach. To my understanding, the difference is that 1) fixed degradation is used in this work, and 2) how the degradation information is embedded and inserted. This is an important discussion for readers to understand the technical novelty of this work. Could you elaborate more on this?

---

> > > > > ### Author Response · Authors · 2023-08-21
> > > > >
> > > > > Thanks for your follow-up questions.
> > > > >
> > > > > `2.a`  We follow the reviewer's advice to provide FID results in ablation studies. Due to the limited time, we only show some representative items to address the reviewer's concerns. Others would be added later.
> > > > >
> > > > > |  Method   | LPIPS  | FID | MACs|
> > > > > |  ----  | ----  | ---- | ---- |
> > > > > |w/ → w/o LayerNorm         | 0.227 | 57.96   |44.1|
> > > > > | w/o Spatial guidance         |0.240 | 68.18 |43.8|
> > > > > |AdaIN                                  |0.240 |  68.24   | 43.9|
> > > > > |DvSR                                  |0.244 | 71.36  | 45.8|
> > > > > |Ours                                   |0.222 |  55.07  | 44.1|
> > > > >
> > > > > As we can see in the table, removing the spatial guidance procedure reduces the FID by more than 10. The ` w/o Spatial guidance ` is slightly better than DvSR, which comes from the design of our basic module (LayerNorm, etc).
> > > > >
> > > > > `2.b`  Our initial motivation is to design the module for the diffusion model to integrate conditional information. So, we did not explore it on the regression task too much. In our model, we only integrate the AKGM module into a shallow Unet architecture. To evaluate it on regression tasks fairly, we integrate the guidance procedure into the convolution-based backbone, which is more straightforward to implement. The computational costs are kept similar by scaling the internal channels.
> > > > >
> > > > > |  Method   | PSNR |
> > > > > |  ----  | ----   |
> > > > > | SID                            |  28.88 |
> > > > > | SID+guidance           |  29.11   |
> > > > > |NAFNet                      |  29.14  |
> > > > > |NAFNet+guidance   |  29.29  |
> > > > >
> > > > > In the above table, we show intermediate results of `+ guidance` models that are trained with $\frac{4}{5}$ full training steps. The guidance procedure improves the regression model by more than 0.15dB, which shows the effectiveness of our method.
> > > > >
> > > > > `3`  **We will try their code on our tasks for further discussion and comparisons.** Some differences:
> > > > >
> > > > > 1. DASR requires a regression loss to learn the degradation information. Our method directly produces a weight map from the initial output.
> > > > >
> > > > > 2. Our method is spatially adaptive according to the image textures. DASR allocates kernels to handle various degradations.
> > > > >
> > > > > 3. Our method is more unified. The degradation information (camera type, ISO, exposure time, etc.) and some other scalar information (scene type, user preference, etc.) can be integrated into our framework.

---

### Official Review · Reviewer_V6gu · 2023-07-05

**Soundness:** 4 excellent
**Presentation:** 4 excellent
**Contribution:** 4 excellent
**Rating:** 7
**Confidence:** 5

**Summary:**

This paper introduces a unified conditional framework for image restoration tasks based on diffusion models. The framework utilizes a UNet to predict initial guidance and incorporates multi-source conditional information into each block to enhance the generative model's guidance. Adaptive Kernel Guidance Block (AKGB) dynamically fuses kernels in each block. Additionally, a patch-splitting strategy is introduced to handle high-resolution images, ensuring the consistent generation of high-resolution images without grid artifacts. Extensive experiments are conducted on extreme low-light denoising, image deblurring, and JPEG restoration.

**Strengths:**

• The setting for generative image restoration is very interesting and should be a promising direction, as it addresses the limitation of existing methods that primarily focus on improving PSNR without consistent visual quality according to human perception. In recent low-level vision research, although PSNR improvements have been observed, the enhancements in visual quality have been minimal. This paper provides a good starting point for generative image restoration.
    • The challenge of maintaining consistency in high-resolution images is important for generative models. The proposed ‘inter-step patch-splitting’ idea is cute and effective to solve this problem for diffusion models.
    • The method demonstrates good generalization across three different image restoration tasks on perceptual quality.
    • The visualization results are impressive. With the perception-oriented target, the method seems to improve the perception quality a lot.

**Weaknesses:**

• The guidance map plays an important role in this method, which output the low-frequency information and serves as the spatial guidance for each block. Visualizing the guidance map can help readers to understand this method better.
    • While the  'Inter-step Patch-splitting Strategy'  is cute, it would be valuable to include quantitative results to demonstrate its effectiveness.
    • There are some mirror typos. Caption of Fig.1; L286 'LayerNomr -> GroupNorm' ; L158 high light;

**Questions:**

• See the weaknesses.
    • It is highly recommended to release the code to encourage further research.

**Limitations:**

In the Limitation and Future Direction section, the authors point out that the method may generate some unnatural textures.

---

> ### Author Rebuttal · Authors · 2023-08-09
>
> > Visualization of the guidance map
>
> We appreciate the reviewer's recognition of our efforts to enhance the perceptual quality and qualitative results of our method. In the revised version, we will include visualizations of the guidance map to provide further insights into our approach.
>
> > It is suggested to add quantitative results for the patch-splitting strategy.
>
> Thank you for your valuable advice. We will include quantitative results around the overlapped regions to show the effectiveness of the patch-splitting strategy.
>
> > Minor issues and code:
>
> Minor issues will be fixed, and the code is preparing.

---

> ### Comment · Area_Chair_MS1c · 2023-08-18
>
> Reviewer V6gu: We need you to respond to the rebuttal ASAP

---

> > ### Comment · Reviewer_V6gu · 2023-08-19
> >
> > After reading the author's response and discussions with other reviewers, my concerns have been well addressed, and the weakness of this paper can be fixed in the final version. I think this paper highlights an important direction to improve the visual quality in low-level vision. Moreover, the visualization results are notably impressive. Consequently, I'd like to change my rating to Accept.

---

### Official Review · Reviewer_8ANp · 2023-07-05

**Soundness:** 2 fair
**Presentation:** 3 good
**Contribution:** 2 fair
**Rating:** 5
**Confidence:** 5

**Summary:**

This paper proposes a novel framework for supervised image restoration. The framework consists of an Initial predictor and a newly designed conditional diffusion model. The initial predictor first produces an initial restoration result, then the conditional diffusion takes the initial result as well as the degradation image as inputs and progressively generates the final result. Besides, the authors also propose a novel solution for restoring high-resolution images. Experiments on low-light denoising, image deblurring, and JPEG restoration show that the proposed framework achieves superior performance in terms of perceptual metrics.

**Strengths:**

1. The solution for high-resolution image restoration seems novel and inspiring.
2. The proposed method achieves superior performance in three restoration tasks in terms of perceptual metrics.
3. The proposed denoiser backbone seems reasonable.
4. The paper is well-written and easy to follow.


**Weaknesses:**

1. Though seems reasonable, there is no evidence that proves the advantage of the designed backbone. At least, the authors should compare it to the denoiser backbone used in SR3 [1]. This is necessary when designing a new backbone instead of using existing ones.
2. It seems that the performance on distortion metrics (i.e., PSNR, SSIM) lags far behind state-of-the-art methods.
3. Some minor issues. (1) The conference information in your citations needs to be updated. For example, SR3 [1], DDNM [2], and LDM [3] should be TPAMI, ICLR, and CVPR, rather than arXiv, CoRR, and None. (2) There exist other methods [3, 4] for high-resolution image restoration, it is appropriate to include them in the discussion.

[1] Saharia et. al., Image super-resolution via iterative refinement. TPAMI 2022.

[2] Wang et. al., Zero-shot image restoration using denoising diffusion null-space model. ICLR 2023.

[3] Rombach et. al., High-resolution image synthesis with latent diffusion models. CVPR 2022.

[4] Wang et. al., Unlimited-size diffusion restoration. CVPRW 2023.

**Questions:**

1. Please see the weaknesses.
2. I wonder if the proposed framework can be used for general image-to-image translation tasks, or benefit the text-to-image task.

**Limitations:**

Given the concerns mentioned in the weaknesses, I give a borderline score. However, I may also change my score if there is enough evidence of the profound value of this paper.

---

> ### Author Rebuttal · Authors · 2023-08-09
>
> > It is suggested to compare the proposed method with the existing backbone.
>
> To demonstrate the advantage of our designed backbone, we included an additional existing method, DvSR [3], in the ablation studies. As indicated in Table 1 of the newly attached PDF, our approach exhibits comparable computational cost to DvSR while achieving significantly better performance in terms of perceptual metrics. The corresponding visual improvements are presented in Figure 1.
>
> > The concern that the performance of the proposed method on distortion metrics (PSNR, SSIM) lags far behind state-of-the-art methods
>
> Note that PSNR and SSIM are actually not directly related to perceptual quality of the restored images. Most previous methods are deterministic and PSNR-oriented, resulting in predicting averaged results with limited visual improvements and higher PSNR, often appearing blurry [4]. Some visualization can be found in Figure 2 of the newly attached PDF.
>
> In contrast, our method learns to predict the potential distribution, which may affect PSNR results but significantly enhances visual quality, yielding sharp and visually appealing outcomes.
>
> > Minor issues.
>
> Thank for your kind comments. We will update the citations and discuss the existing latent space [1] and Mask-Shift [2] solutions.
> It’s possible to extend to other image-to-image translation tasks (colorization, deraining, etc). We are actively working on incorporating these tasks and will present them in the final version.
>
> [1] Rombach et. al., High-resolution image synthesis with latent diffusion models. CVPR 2022.
>
> [2] Wang et. al., Unlimited-size diffusion restoration. CVPRW 2023.
>
> [3] Whang et. al., Deblurring via stochastic refinement. CVPR, 2022.
>
> [4] Sajjadi et. al., EnhanceNet: Single Image Super-Resolution Through Automated Texture Synthesis. ICCV 2017.

---

> > ### Comment · Reviewer_8ANp · 2023-08-16
> >
> > Thanks for your response. I will keep my rating.

---

> ### Comment · Area_Chair_MS1c · 2023-08-18
>
> Reviewer 8ANp: Your review does not justify your recommendation to (borderline accept). Your response to the authors rebuttal is not informative. Unless there is further substantive discussion, I will ignore this review.

---

### Official Review · Reviewer_4fFj · 2023-07-07

**Soundness:** 3 good
**Presentation:** 3 good
**Contribution:** 3 good
**Rating:** 5
**Confidence:** 3

**Summary:**

This paper proposes a new framework for image restoration with diffusion models. They design strategies to better use the condition information and also propose a strategy for high-solution images. The results are competitive over baselines. The presentation is clear. I tend to accept this paper.

**Strengths:**

- The performance is strong. It outperforms existing baselines on several tasks and datasets quantitatively. The perceptual performance is also good. The data fidelity and details are good.
- This paper proposes detailed experiments and analysis for each design of the framework, which helps the readers to understand their method. However, I would say that it would be better to provide some results on high-resolution images.
- The paper writing is good and clear.
- Most results are obtained on real-world datasets.

**Weaknesses:**

- It would be better to provide some results on high-resolution images.
- It would be better to provide results on more image restoration tasks such as colorization, inpainting.

**Questions:**

See weakness.

**Limitations:**

See weakness.

---

> ### Author Rebuttal · Authors · 2023-08-09
>
> > Provide more high-resolution image results
>
> In our paper, the testing images of SID dataset are high-resolution (4256x2848). Due to the limited space, we only show some crops in fig.3 and provide some full-resolution images in the supplementary material. We will revise paper to make this point clear.
>
> > Extension to more image restoration tasks such as colorization and inpainting
>
> It’s possible to extend the method to other img2img tasks like colorization, deraining, etc. We are actively working on incorporating these tasks and will present them in the final version.

---

> > ### Comment · Reviewer_4fFj · 2023-08-16
> > **Keep my rating as borderline accept**
> >
> > Thanks for your response. I will keep my rating as borderline accept.

---

> ### Comment · Area_Chair_MS1c · 2023-08-18
>
> Reviewer 4fFj: Your review says Soundness: 2 fair, Presentation: 2 fair, Contribution: 2 fair, but you recommended borderline accept. This doesn't make much sense. The rebuttal and your response to it are also uninformative. Unless there is additional meaningful discussion, I will be forced to ignore this review.

---

> > ### Comment · Reviewer_4fFj · 2023-08-19
> > **More details**
> >
> > It solves my concerns.
> >
> > 1) It argues that "A simple yet effective inter-step patch-splitting strategy is proposed for handling high76 resolution images. This practical strategy enables diffusion models to generate consistent **high-resolution** images without grid artifacts."  However, I did not notice these results. Hence, I ask the authors to provide some results. They explain the reasons and I buy their explanations.
> >
> > 2) I hope results on different tasks can be provided. They agree and promise that they will update it in the final version.

---

### Author Rebuttal · Authors · 2023-08-09

We thank reviewers for their valuable and professional comments. Overall, reviewers (4fFj, 8ANp, V6gu, 37QZ) acknowledge the novelty and the performance of our paper. We have uploaded a one-page PDF containing figures and tables to help us addressing the reviewers' concerns, and then, we will respond to each reviewer individually.

---

### Decision · Program_Chairs · 2023-09-21

**Decision:**

Accept (poster)

**Comment:**

Despite one of the reviewers continuing to have strong strong reservations on lack of novelty (in the direction of super-resolution), the preponderance of opinion based on the reviews and rebuttals is that the paper is worthy of publication. I recommend that the paper be accepted for poster presentation at NeurIPS 2023. The paper makes a good contribution to the use of conditional diffusion models for image restoration by proposing a unified conditioning framework. The paper is generally well-written and I find the experiments and rebuttals convincing.